# Role of Extracellular Matrix and Inflammation in Abdominal Aortic Aneurysm

**DOI:** 10.3390/ijms231911078

**Published:** 2022-09-21

**Authors:** Karolina L. Stepien, Karolina Bajdak-Rusinek, Agnieszka Fus-Kujawa, Wacław Kuczmik, Katarzyna Gawron

**Affiliations:** 1Department of Molecular Biology, Faculty of Medical Sciences in Katowice, Medical University of Silesia, Katowice, Medykow 18 Street, 40-752 Katowice, Poland; 2Department of Medical Genetics, Faculty of Medical Sciences in Katowice, Medical University of Silesia, Katowice, Medykow 18 Street, 40-752 Katowice, Poland; 3Department of General, Vascular Surgery, Angiology and Phlebology, Medical University of Silesia, Katowice, Ziolowa 45/47 Street, 40-635 Katowice, Poland

**Keywords:** abdominal aortic aneurysm, extracellular matrix, inflammatory process, biomarkers, Hcy, OPN, OPG, Cat

## Abstract

Abdominal aortic aneurysm (AAA) is one of the most dangerous cardiovascular diseases, occurring mainly in men over the age of 55 years. As it is asymptomatic, patients are diagnosed very late, usually when they suffer pain in the abdominal cavity. The late detection of AAA contributes to the high mortality rate. Many environmental, genetic, and molecular factors contribute to the development and subsequent rupture of AAA. Inflammation, apoptosis of smooth muscle cells, and degradation of the extracellular matrix in the AAA wall are believed to be the major molecular processes underlying AAA formation. Until now, no pharmacological treatment has been implemented to prevent the formation of AAA or to cure the disease. Therefore, it is important that patients are diagnosed at a very early stage of the disease. Biomarkers contribute to the assessment of the concentration level, which will help to determine the level and rate of AAA development. The potential biomarkers today include homocysteine, cathepsins, osteopontin, and osteoprotegerin. In this review, we describe the major aspects of molecular processes that take place in the aortic wall during AAA formation. In addition, biomarkers, the monitoring of which will contribute to the prompt diagnosis of AAA patients over the age of 55 years, are described.

## 1. Introduction

Abdominal aortic aneurysm (AAA) is an asymptomatic disease resulting from the local expansion of the aorta to more than 50% of its normal diameter. AAA can also be defined as a dilation of the aortic wall where the diameter of the aorta is greater than or equal to 3.0 cm [1]. The enlarged diameter of the aorta is capable of rupture if it is undetected and left untreated. This disease mainly affects men over 55 years of age [2,3]. Results of a population screening revealed that, at the age of 55 years, 4 to 7% of men and 1 to 2% of women are predisposed to the aneurysm [2,4]. Despite the fact that AAA is predominant in men, the rate of rupture of small aneurysms is really three times higher in women [5]. It has been proven that, in women, the medium diameter of the aorta undergoing rupture is 5 cm, whereas in men it is 6 cm. Moreover, in aneurysms 5 to 6 cm, the risk of rupture is 4 times higher in females than in males. The reason is the smaller diameter of the healthy female aorta [5]. A ruptured aneurysm significantly affects the health and life of patients, with the survival rate of ruptured aneurysms only from 10 to 20% [6,7,8].

AAA is a multifactorial disease; thus, it is impossible to identify a single factor which promotes the formation of the aneurysm. Nevertheless, among the most important factors for a higher risk of developing aneurysm are cigarette smoking, male gender, age, family history, hypertension, and hypercholesterolemia [1,2,4]. Additionally, apoptosis of smooth muscle cells (SMCs), excessive degradation of extracellular matrix (ECM), inflammation, and oxidative stress likewise negatively affect the functioning of the aorta [1,2,3,4].

This review describes selected molecular processes that take place in the aortic wall during the formation of an abdominal aortic aneurysm. Additionally, we focus on discussing AAA biomarkers, as their monitoring in patients over 55 years of age will contribute to early AAA diagnosis.

## 2. The Aortic Wall Structure

The circulatory system, which integrates structurally and functionally in our body, consists of two major components, the blood vessels (arteries and veins) and the heart. Blood vessels as tubes form pathways for blood flow, whereas the heart is a muscle that, by pumping the blood, keeps it moving. The arteries differ due to their anatomical structure and physiological properties, forming aortas, main arteries, arterioles, and arterial capillaries and, subsequently, veins with increasing diameters when close to the heart. The human aorta is the largest conducting artery, whose primary function is blood transport and flow equalization. The aorta is an artery of the elastic type, which consists of three main layers: tunica intima, tunica media, and tunica adventitia (Figure 1). Furthermore, the tunica adventitia is separated from the tunica media by the external elastic lamina, and the tunica intima is separated from the tunica media by the internal elastic lamina. The major cellular components of the aorta wall are endothelial cells (ECs), smooth muscle cells (SMCs), and fibroblasts (FBs). Additionally, the wall is made of ECM proteins specific for each layer (Table 1). 

### 2.1. Fiber Type Structures Determine the Wall Properties

The biologically active and main component of blood vessels is the extracellular matrix, which further provides support for the cells. The ECM is a very dynamic structure which plays a key role in regulating vascular function under both physiological and pathological conditions. The proper functioning of the ECM allows for the maintenance of the aortic homeostasis and mechanical properties. The main macromolecules forming the ECM in aorta and other large arteries are glycoproteins, proteoglycans, collagens, and elastin [11]. ECs specific for aorta are present in the tunica intima and rest on the basal lamina. The basal lamina is a type of basement membrane with a mesh-like structure resulting from properties of its major components such as non-fibrillar collagen, laminins, and fibronectin. The smooth muscle cells specific to the aortic wall are found in the middle layer of the aorta and are responsible for the production of proteins such as elastin and collagens, which build the ECM in this part of the aorta. While, in the adventitia layer, collagens, osteopontin, and fibronectin are produced by fibroblasts specific for this part of the aorta wall [11].

#### 2.1.1. Types of Fiber Structures

Both the strength and the elasticity of the aorta wall result from the supramolecular structure of respective elements of ECM, mainly elastin and fibrillar collagens. Elastin is formed by the polymerization of tropoelastin monomers and allows for the expansion and affects the flexibility of the vessels. Collagens self-assembling into large, highly ordered fibrils additionally reinforced with cross-links protect the vessel against tangible stretching. Both proteins directly interact with various integrins and other proteins to facilitate the adhesion, proliferation, and migration of SMCs.

In the middle layer of the aortic wall, elastin fibers are most abundant. The largest component of elastic fibers expressed most rapidly in the vascular smooth muscle cells (VSMC) in the aorta. Elastin fibers are peripherally bound with SMCs and, additionally, with collagen fibers. This structure is a lamellar unit. In development, the number of units is fixed and coincides with the tension in the aortic wall. During low tension, the laminae are wavy, while at increased concentration, the laminae stretch and straighten. Thus, along with collagen fibers, elastin is the main protein responsible for elasticity. Arteries are constantly exposed to high blood pressure. Elastin fibers and fibrils form the strong scaffold of the aorta, thus protecting the aorta against mechanical damage [11,12,13].

The second most important fibers are collagen fibers. They are arranged between SMCs and elastic lamellae and interact with the main vascular cell types, specifically FBs, SMCs, and ECs. Collagens also play a crucial role in various biological processes such as tissue remodeling and repair, maintenance of tissue strength, interactions with cell specific receptors affecting cell adhesion, differentiation, growth, and survival, as well as various pathological processes [11]. In the non-pathological aorta, collagen type I and III are the fibril forming collagens in the adventitia, media, and intima layers, whereas major collagen found in the ECs and SMCs are non-fibrillar collagen types IV and V. In AAA, collagen type I and IV are present in the media and intima layers, while type III was found in the adventitia layer. Collagen type III forms copolymers with collagen type I, and its fraction determines the higher elasticity of the aorta’s wall, while collagen type I provides more resistance to the mechanical properties of the aorta’s wall [11,14].

#### 2.1.2. Modifications of ECM Structural Elements

The main macromolecules forming ECM undergo posttranslational modification, both intra- and extracellularly. A variety of chemicals can be added to peptide chains of elastin and collagens. The modifying molecules may be of different types, such as proteoglycans (PGs) and glycoproteins.

Proteoglycans and glycoproteins are important compounds of the extracellular matrix, comprising the non-fibrillar fraction of the ECM. These molecules have complex structures that contain a core protein with covalently attached chains of glycosaminoglycans (GAGs). They provide a coherent matrix structure, connecting molecules to each other as well as to the ECM. They have several domains binding various molecules which form a cross-linked network. In the aortic wall large proteoglycans such as versicans mainly are found, but also small leucine rich proteoglycans (SLRP) such as decorin (DCN), lumican (LUM), and biglycan. These proteins form the pericellular ECM and are involved in the promotion of proliferation and migration of vascular smooth muscle cells (VSMCs) [9].

Decorin is expressed by a large group of cells, for instance, SMCs and FBs [9]. This small proteoglycan contributes to the maintenance of matrix homeostasis by the regulation of collagen fibrillogenesis and its degradation, cell signaling, and cell growth. Decorin is found in association with collagen fibers, thus, it acts as an anchor between the collagen fibers, affecting their elasticity and stretching. DCN is also involved in the formation, fusion, and organization of fibers during collagen fibrillogenesis [9,15]. DCN and LUM are involved in the process of collagen fibrillogenesis in connective tissue in humans as well as in animal models (LUM knock-out mice) [16]. LUM binds to fibrillar collagens, regulating their thickness and the spacing between them, which is important for tissue integrity [16]. Moreover, LUM affects cell growth, apoptosis, migration, invasion, immunoresponse, and angiogenesis [17].

Biglycan is an essential component of ECM, interacting with proteins such as collagen and elastin, which contributes to maintaining the correct cross-linking of the extracellular matrix. This proteoglycan regulates collagen fibrillogenesis and is expressed throughout a human’s aortic wall. In response to tissue damage, the soluble form of biglycan is produced by ECM proteolysis or by de novo synthesis in macrophages [9,18].

Versican is a large chondroitin sulfate (CS) proteoglycan which is a key component of the ECM. Versican appears in the intimal and medial layers of the aorta. Five different isoforms of versican: V0, V1, V2, V3, and V4, have been reported [9,10,12]. Specifically, isoform V1 supports proliferation and inhibits apoptosis, whereas isoform V2 exhibits antiproliferative activity. On the other hand, isoform V3 regulates matrix formation and inhibits cell migration and proliferation. Versican is synthesized in epithelial, endothelial, and stromal cells, and its production is regulated by proinflammatory cytokines and growth factors. Moreover, versican is a reservoir of cytokines and chemokines, for example, stromal cell-derived factor-1β (SDF-1β). It regulates their release when necessary, in order to precisely control the cells’ activity and their behavior. Another essential role is participation in the inflammatory process caused by infection or tissue damage [9,19,20].

Fibronectin (FN) is the glycoprotein commonly present in the wall of the aorta. It is a multidomain and multifactorial glycoprotein, which, in the ECM, plays a structural and regulatory role by controlling interactions between cells and matrix. FN mainly affects the cell migration, differentiation, angiogenesis, growth, and repair process. It interacts with integrins participating in the transmission of signals from the external environment to the interior of the cell [9,21]. Fibronectin determines the incorporation into the ECM of such proteins as fibrillin, collagens, fibulin, latent TGF-β binding protein (LTBP), and tenastin-C. It is known that some of the proteins bind directly to FN, while some of them use FN as a scaffold for the independent insertion fibers [22].

### 2.2. ECM Remodeling: The Major Players in This Process

Although ECM in blood arteries, particularly in large vessels such as aorta, is stable for tens of years, it is subjected to limited remodeling due to physiological processes such as growth of the body and adaptation of the organism to individual physical activity. More intense changes in the structure of the aorta wall’s ECM occur following exposure to excessive physical force, such as high blood pressure, damage of the epithelium by toxic agents and other abnormalities affecting the wall under the influence of metabolites such as cholesterol, triglycerides, metal ions, etc. The major factors involved in remodeling of the ECM in the aortal wall are metalloendopeptidases, both those building the ECM structure, e.g., procollagen pro-peptides (BMP1 and ADAMTS2) and those degrading the ECM structure, e.g., proteinases mainly classified as matrix metalloproteinases (MMPs) (Table 2). BMP1 and ADAMTS2 are zinc/calcium ions dependent on metalloproteinase converting procollagens to self-assembling collagen monomers. They catalyze removal of C- and N-pro-peptides, respectively, from the central triple helical collagen domains. Their action is critical for the production of collagen fibrils in connective tissue. In aortal walls, the lack of their activity leads to weaker wall structure, possible development of aneurysm and its rupture, among other general tissue problems [23,24,25]. Genetic, inherited diseases manifested by the development of aneurysms are some cases of progressively deforming type 3 osteogenesis imperfecta (OI) with a mutation in the gene encoding BMP1, [26] also classified as OI type XIII [27], dermatosparaxis with mutations in ADAMTS-2 [28], and mutations in COL3A1, causing the vascular type of Ehlers–Danlos Syndrome [29]. These two types of metalloendopeptidases constitute a class of highly homologous enzymes with zinc ion (Zn^2+^) in the active centers of their catalytic domains and with multidomain structures [30,31,32,33].

MMPs, through ECM modifications, facilitate cell proliferation, migration, and differentiation. Furthermore, they are involved in angiogenesis, cell apoptosis, and tissue repair. These endoproteinases also play a role in various signaling pathways. Increased expression and activity of MMPs occur during normal biological processes such as wound healing, but also in pathologies, especially in cardiovascular diseases or aneurysm formation such as in AAA.

MMPs are produced by different types of cells and tissues. These enzymes are secreted by both ECM forming cells such as FBs, ECs, and VSMCs and proinflammatory cells, e.g. macrophages and neutrophils [23]. Out of the 28 described MMPs, 23 were found in humans. They are classified to six different families, depending on the specific substrate of the ECM component [34,35]. On the basis of the substrates and the organization of the domains of metalloproteinases, the following families of these enzymes have been proposed: collagenases, gelatinases, stromelysins, matrilysins, metalloelastase, and membrane-type (MT) MMPs (Table 2) [4,23,24].

It is crucial to control the MMPs’ concentrations and the activities, because the elevated activity of these enzymes results in excessive ECM degradation, which also contributes to the formation of an AAA [36,37,38]. One of the controlling systems includes tissue inhibitors of metalloproteinases (TIMPs). They are responsible for the regulation of the correct level of MMP activity [34]. Four TIMPs have been identified and characterized, namely TIMP-1, TIMP-2, TIMP-3, and TIMP-4. All of them are able to inhibit the proteolytic activity of all known MMPs. The inhibitory performance varies with the type of tissue inhibitor. TIMPs bind to MMPs through a covalent bond and prevent the substrate from accessing the catalytic site [38]. The balance between TIMPs’ and MMPs’ activity influences the normal processes related to tissue remodeling, angiogenesis, or tumor formation. The activity of TIMPs through the inhibition of MMPs determines the balance between the synthesis and proteolytic degradation of ECMs. The interaction of MMP and TIMP does not only concern the decomposition of the ECM. It is a complex process involving biologically active proteins such as cytokines, chemokines, and cell surface proteins. Regulation of biologically active proteins with the participation of MMP/TIMP complexes has an indirect effect on matrix remodeling. The role of TIMP in regulating ECM remodeling depends on a specific metalloproteinase inhibited by specific TIMP in a tissue. Additionally, the TIMP family is involved in reducing the degradation of the ECM by interacting with a variety of MMPs [2,34,37,38].

**Table 2 ijms-23-11078-t002:** Classification of selected metalloproteinase crucial for ECM remodeling and their roles in AAA pathogenesis.

Family	Metalloproteinase	Source	ECM-Substrate	Potential Role in AAA Pathogenesis	References
**Tolloids**	**BMP-1** bone morphogenetic protein 1 (BMP-1)	osteoblasts chondroblasts	Procollagens types I and III, Laminin 5 gamma 2 chain	Converting procollagens to collagens	[39]
	**mTLD** mammalian Tolloid	ECM	Procollagen types I, II, and III	Converting procollagens to collagens	[33,40]
	**TLL-1** Tolloid-like protein 1		Chordin, Pro-lysyl Oxidase	Crucial for activation of BMP2 and BMP4 for cell differentiation, critical for providing active Pro-lysyl Oxidase to crosslink collagen monomers and collagen fibrils	[41,42]
	**TLL-2** Tolloid-like protein 2		Pro-lysyl Oxidase	Critical for providing active Pro-lysyl Oxidase to crosslink collagen monomers and collagen fibrils	[43]
**ADAMTS**	**ADAMTS-2** A disintegrin and metalloproteinase with thrombospondin motifs 2	fibroblasts	Procollagen types I and III	Converting procollagens to pC collagens	[44]
**Collagenasses**	**Collagenase-1: MMP-1**	ECs, SMCs, FBs, macrophages, platelets,	Collagen triple helix, Versican, aggrecan, nidogen, perlacan, proteoglycan link protein	Development of the inflammatory process in the aortic wall; modulate the process of aortic rupture and dissection	[4,34,35]
	**Collagenase-2: MMP-8**	Macrophages, neutrophils,	Collagen triple helix; elastin, fibronectin, laminin, aggrecan	Significant expression in expanded and rupture AAA	[34,35]
	**Collagenase-3: MMP-13**	VSMCs, macrophages,	Collagen triple helix; gelatin, fibronectin, laminin, tenascic	Significant expression in AAA: symptomatic and ruptured AAA	[4,34,35]
**Gelatinases**	**Gelatinase-A: MMP-2**	ECs, vascular smooth muscle (VSM), SMCs, FBs, macrophages, platelets, leukocytes, adventitia	Collagen, triple helix; gelatin, elastin, aggrecan, fibronectin, versican, proteoglycan link protein	Elevated levels in developing aneurysms	[34,35]
	**Gelatinase-B: MMP-9**	ECs, VSM, platelets, macrophages, adventitia	Collagen: IV, V, VII, X, XIV, gelatin, elastin, aggrecan, fibronectin, laminin, versican, proteoglycan link protein	High levels in developing aneurysms; stimulates the inflammatory response in AAA	[10,34]
**Stromelysins**	**Stromelysin-1:** **MMP-3**	ECs, FBs VSM, intima, epithelium	Collagen: II, III, IV, IX, X, XI, gelatin, aggrecan, decorin, elastin, fibronectin, versican, laminin, proteoglycan link protein	Promotes AAA	[4,34,35]
	**Stromelysin-2: MMP-10**	ECs, FBs	Elastin, fibronectin, gelatin I, link protein, casein, fibronectin	High level in atherosclerosis	[34]
**Matrilysins**	**Matrilysins- 1: MMP-7**	ECs, VSM, intima	Collagen: IV, X, gelatin, aggrecan, elastin, fibronectin, laminin, proteoglycan link protein, N-cadherin	Increase expression in AAA	[34]
**Metalloelastase**	**MMP-12**	Macrophages, SMCs, FBs	Collagen IV, gelatin, elastin, fibronectin, laminin	Enhance AAA formation	[34,35]
**MEMBRANE TYPE MMPs**	**MT1-MMP: MMP-14**	FBs, VSM, SMCs platelets, macrophages	Collagen I, II, III, gelatin, aggrecan, elastin, fibronectin, laminin, proteoglycan, vitronectin	Direct degradation of ECM in the tunica media and adventitia in aortic wall—formation of AAA	[34,35]
	**MT1-MMP MMP-15**	FBs, leukocytes	Collagen type I, gelatin, aggrecan, fibronectin, laminin, nidogen, tenascin, perlacan	Reduction in cell adhesion	[34,37]
	**MT1-MMP MMP-17**	VSMCs	Osteopontin in VSMCs, gelatin, fibrin	Restrain AAA formation	[34]

## 3. The Pathogenesis of the AAA

### 3.1. ECM Degradation

The pathogenesis of AAA is a multistage process involving several biological steps and risk factors. It seems that no specific individual molecular mechanism is responsible for the formation of an AAA. It is known that the degradation of the ECM negatively affects the functioning of the aorta. The ECM is a skeleton of the aortic wall responsible for the transmission of muscle fiber forces and for its maintenance and repair. Abnormal remodeling of the vessel wall under the pathological mechanical forces exerted by blood pressure adversely affects the expansion of the vessel wall, contributes to the formation of the AAA, or leads to rupture.

There is an anatomical and functional heterogeneity that determines the diversified pathogenesis and, therefore, potential differences at the molecular level. The attached photos show fragments of aneurysms recovered upon surgical treatment of three different patients. Morphological differences between patients are evident in terms of size and morphological appearance (Figure 2).

The most important cause of AAA formation is ECM degradation by MMPs. These enzymes are involved in changing specific regions of the aorta, such as the tunica intima and media. They mainly contribute to: calcification, thrombosis, appearance of lipids in foam cells, adventitial inflammatory infiltrate, and rupture of the aorta layer. Tunica media degradation is one of the most common causes of AAA. In addition, the loss of the basic proteins of the ECM, which are collagen and elastin, also contributes to the degradation of the aorta wall structure. Table 2 shows a number of metalloproteinases that contribute significantly to the formation of AAA. It has been shown that MMP-1, -2, -3, -9, -12, -13, and -14 achieve high levels of expression in patients with AAA [4,34,35]. Among the listed metalloproteinases, MMP-9 accomplished the highest level of expression. Plasma levels of MMP-9 are in the range of 0.06–0.6 µg/mL. High levels of MMP may occur in patients where the aneurysm is very large and can rupture. There is also a correlation between the level of MMP concentration and the stage of development and the size of the AAA. It turns out that the plasma level of MMP-9 is also directly related to the level of MMP in tissue. Essential MMP-9 levels drop significantly after the aneurysm’s surgical treatment [4,34]. Furthermore, high levels of MMP-9 in plasma are closely related to atherosclerosis and acute coronary syndrome, but also to aneurysm formation and rupture. Additionally, an increased plasma concentration of MMP-9 is significantly associated with a greater aortic wall thickness and a larger aortic lumen diameter, but not with a higher ratio of the aortic lumen diameter to aortic wall thickness [45].

### 3.2. Inflammatory Process

The inflammatory process that arises in the AAA wall is a key factor in the formation of an aneurysm. The inflammation involves a number of cells of the immune system. The most important cells are: lymphocytes, macrophages, and mast cells. These cells penetrate into the tissues through the layers of the aorta: intima, media, and adventitia. The cells of the immune system that participate in the immune response are shown in Figure 3.

#### 3.2.1. Lymphocyte

The relevant group of cells involved in the inflammatory process of the aortic wall is T and B lymphocytes [4,46]. T lymphocytes are a very large population of cells with a diverse classification system and physiological functions. They are first classified based on the surface expression of CD8 and CD4 molecules. The dominant and most important cells in the aneurysm wall are CD4^+^ T cells. They secrete a series of cytokines that control the dynamic metabolism of the ECM through macrophage enrolment, ECM regulation, and the synthesis of proteolytic enzymes. Secondly, CD4^+^ T cells can be divided into helper (Th), effector (T_eff_), and regulatory (T_reg_) lymphocytes. Among them, there are subsets: Th1, Th2, and Th17. The classification and division of lymphocytes result from the cytokines required for stimulation along with the secreted products and their physiological functions [47]. Th1 are activated by interleukin 12 and act through the STAT4 and T-bet signaling pathway to secrete IFN-γ, TNF-α, and TNF-β. This influences the activation of macrophages and then activates the intrinsic pathway, enhances Th1 development, and inhibits the alternating differentiation of T lymphocytes. In this cycle, INF-γ activates macrophages by increasing the expression of cytokines, chemokines, and adhesion molecules. In this case, macrophages produce Il-12, which positively influences the further activation of macrophages. The consequence of this process is the degradation of the ECM, which significantly promotes the formation of AAA. There is a relationship in the level of expression between IFN-γ and Th2. High levels of IFN-γ were found in the tissues, but the high expression of Il-4 that is characteristic of Th2 was not found. High levels of IFN-y were also found in the blood serum of patients with AAA. Importantly, this high level was commensurate with the rate of the aneurysm’s growth [47].

Th2 is the second important group of cells involved in the inflammatory process of AAA. These cells are considered anti-inflammatory cells. Interleukin 4 is a factor that stimulates these cells; it differentiates CD4+T cells into cells with the Th2 cells phenotype. This process occurs via signaling pathways such as STAT6 and GATA3. The Th2 cell secretes several interleukins: Il-4, Il-5, Il-10, and Il-13. The function of these interleukins is to reduce the cytoxicity of macrophages and reduce the expression of proinflammatory mediators and MMPs. The development of anti-inflammatory macrophages with the M2 phenotype enhances Il-13 and is also responsible for the increased expression of MMPs [47].

Research proves that in tissue sections of aneurysms, the number of B and T lymphocytes is significantly increased. It is important that the density of lymphocytes present in the tissues negatively correlates with the content of collagen and elastin. This is due to adaptive involvement of the immune system cells in the instability of the aneurysm. Cells such as Th1, Th2, CD4^+^, and CD8^+^T take part in the aneurysm initiation process. It is obvious that these cells are present in the wall of the AAA, while less is known about the number, exact location, and degree of these cells’ activation in the pathological aorta. The presence of lymphocytes in the aortic wall and no association/correlation of these cells with the size of the aneurysm have been demonstrated [46]. It is worth mentioning that regulatory T lymphocytes inhibit or weaken the process of aneurysm formation. This is because regulatory lymphocytes secrete interleukin 10 (Il-10), which has anti-inflammatory properties, and TGF- β, that has stabilizing properties for the aneurysm [4].

In studies performed in mice and humans [47], it has been found that Th2 cells are the main cells of the immune system that contribute to the formation of AAA [47]. It has been shown that the tissues of the aneurysm have a high level of cytokines secreted by Th2 cells, while a low expression of cytokines related to Th1 cells, mainly IFN-γ, was observed in human tissue [47]. It was also found, in human AAA, that Il-4 overexpression probably inhibits Th1 differentiation, while Il-5 enhances elastolytic activity [47]. It has been shown that, in a mice aortic allograft, the development of AAA was enhanced in mice deficient in IFN-γ, while in mice deficient in IL-4, no AAA development was observed [47,48,49]. In contrast to the data obtained from mouse studies, Il-4 positively influences the production of ECM proteins in human fibroblasts and, additionally, inhibits the expression of MMP-1 and MMP-9 in alveolar macrophages [47].

A consecutive class of cells belonging to the helper lymphocytes, Th17, plays a role in the formation of the aneurysm. These cells differ from the previously described Th1 and Th2 cells. Differences appear in the stimulating factors, the factors secreted by these cells, and the signaling pathways in which they participate. Th17 is stimulated by Il-23, Il-1, and Il-6, which are responsible for the induction of the orphan receptor associated with retinoid acid. Orphan receptors γt (RORγt) and STAT3 contribute to the secretion of Il-17. Importantly, Il-17 has six isoforms, while Th17 cells secrete only isoform A and F. Il-17 is important in the production of peroxides in the vessels, mainly mediating inflammatory diseases. Data obtained from studies on murine models have shown that the deficiency of Il-17 limits macrophages in the vessels, while high expression of Il-17 promotes the presence of macrophages in the aortic wall [47,50,51].

Regulatory lymphocytes (T_reg_) are the last lymphocyte population involved in the process of AAA formation. T_reg_ are a special subclass of CD4^+^ T cells; they are responsible for immune tolerance and negatively influence the proinflammatory effects of the effector subclasses of T. Regulatory lymphocyte stimulating factors include Il-2 and TGF-β. T_reg_ stimulating factors include Il-2 and TGF-β. These factors stimulate T_reg_ to produce Il-10 and TGF-β through the STAT5 and Foxp3 (forkhead box P3) signaling pathways. The functions of T_reg_ are crucial for many biochemical processes of ECM, but not exclusively. One of the most important functions is to halt the inflammatory process, limit the damage, and start the matrix repair process. The inhibition of the inflammatory cascade is mainly achieved by blocking the secretion of TNF-α and IFN-γ from the effector cells. These cells are also responsible for the destruction of the autoreactive forms of T cells that appear through the formation by products during matrix degeneration. Another important function is the inhibition of T_eff_ cell proliferation. If T_eff_ cell proliferation is not inhibited, control of the inflammatory response is impaired and out of control. Research shows that patients with AAA have noticeable deficiencies in the T_reg_ cell population [47].

In conclusion, generally, CD4^+^ cells belong to T effector cells, whose role is to regulate the acute inflammatory response. During acute trauma, they participate in the host’s standard defense. In the initial phase of the immune response, T_eff_ cells secrete and induce the secretion of proteases, thanks to which inflammatory cells can migrate to damaged tissues to contact the damaging factor. In a normal process, the T_reg_ cells enable the response to be resolved. However, in the case of the inflammatory process that occurs in AAA, the appropriate anti-inflammatory mechanisms do not work, which results in progressive degradation of the matrix [47].

#### 3.2.2. Macrophages

The most common leukocytes in the media and adventitia of the aorta are macrophages. These cells are responsible for the initiation and adaptation of the immune response. Macrophages penetrate the aortic wall, secreting molecules such as TGF-β and Il-6 that directly contribute to the degradation of ECM, which significantly promotes the formation of AAA. Two groups of macrophages play a major role in the formation of aneurysm macrophages with the phenotype M1 and M2 [52,53]. Macrophages have different functions: M1 macrophages are proinflammatory, whereas M2 macrophages do not demonstrate proinflammatory properties—they rebuild and repair ECM [33]. Macrophages with the M2 phenotype, thanks to the secretion of IL-10 and profibrotic factors, which are TGF-β, achieve the anti-inflammatory effect. TGF-β is an essential factor because its protective effect controls the excessive activation of monocytes/macrophages and the lack of monocytes inhibits the development of AAA. The regulation of the chronic inflammatory process is mediated by polarization of the macrophage M1 and M2 populations. Penetrating the macrophage, M2 can transform into M1 and vice versa [52,53].

The most important cytokines modulating the infiltration of macrophages in the inflammatory process are: granulocyte–macrophage colony stimulating factor (GM–CSF), monocyte chemotactic protein-1 (MCP-1), interleukin-6 (Il-6), interleukin 23 (Il-23), and TGF-β (transforming growth factor-β) [53,54,55]. GM–CSF gene expression is related to the number of macrophages present in the aorta. It is worth adding that the increasing secretion of GM–CSF may contribute to the formation of AAA. Scientists argue that GM–CSF is a key regulatory factor of AAA. It has been shown that the infiltration of macrophages and the secretion of MMP-9 will be reduced if the GM–CSF pathway is blocked [56].

The second factor, MCP-1, is a chemokine, and it is released by inflammatory cells and by endothelial cells. Along with macrophages, it penetrates the aortic wall and contributes to the formation of AAA. Disturbances in monocyte chemotactic protein-1 secretion appear before the immune response. SMCs secrete MCP-1, while this chemokine affects MMP-9 by increasing its synthesis. This correlation significantly contributes to the development of AAA. A characteristic property of SMCs’ apoptosis is that they attract monocytes and other leukocytes by secreting MCP-1. Notwithstanding, MCP-1 stimulated macrophages will continue to induce apoptosis in SMCs. MCP-1 enhances the migration of macrophages and their cytotoxicity, which changes the phenotype of SMCs and leads to their apoptosis [53].

Another significant cytokine is interleukin 6 (Il-6), which is a proinflammatory cytokine that affects the apoptosis of SMCs and enhances the remodeling of the ECM by MMPs. It has been shown that elevated levels of Il-6 occur in patients with AAA [55,57].

TGF-β is another noteworthy cytokine. This factor controls the process of cell growth, differentiation, proliferation, and cell apoptosis. TGF-β is also involved in controlling the immune system, thus influencing the process of AAA. For transforming growth factor-β isoforms: TGF-β1, TGF-β2, and TGF-β3, adequately, there are three classes of receptors (TβR): TβRI, TβRII, and TβRIII. Activation of this cytokine can occur through various signaling pathways, for example, via the Smad pathway, MAPK pathway, or PI3K pathway. Studies show that there is reduced signaling by TGF-β in AAA. The loss of one copy of exon 8 in TβRII was associated with a reduced level of TβRII expression [58].

Additionally, TGF-β also has a protective effect and inhibits the inflammatory process, which was proved in murine models. Researchers administered antibodies that neutralized TGF-β. This resulted in an infiltration of macrophages and monocytes in animal models. Thus, TGF-β may reduce inflammatory cell recruitment and the potential release of proteolytic enzymes that degrade the matrix [58].

#### 3.2.3. Mast Cells

In tissues obtained from patients with AAA, the presence of mast cells was observed. Mast cells are proinflammatory cells that play a role in allergic reactions, atherosclerosis, and AAA development. They participate in the immune response as a result of the resulting inflammatory process. It causes the production of secretory molecules such as inflammatory factors and extracellular proteases. Mast cells play a key role in host defense against pathogens by secreting mediators. It was found that mast cells contribute to a significant increase in the diameter of AAA, the degradation of elastic tissue, and the inflammatory process of the adventitia. Therefore, the mast cell population is greater in the adventitia and media layer of the aorta. Moreover, the number of degranulated mast cells is also significant. In addition, the expression of tryptase and SCF, characteristic of mast cells, is higher in AAA patients compared to patients with arthrosclerosis [59]

The inflammatory process, mainly in the adventitia layer, is an indispensable element contributing to the formation of AAA. Mast cells are able to be active T lymphocytes and macrophages due to the secretion of appropriate proinflammatory mediators and cytokines. On the other hand, mast cells can be activated by T lymphocytes through direct contact. It is worth adding that mast cells produce tryptase and chymase, which contribute to the degradation of ECM by activating MMPs, the apoptosis of SMCs, which is the basis for AAA formation [59,60].

Mast cells initiate the inflammatory response by releasing inflammatory cytokines, growth factors, and proteases. The proinflammatory cytokines are mainly IFN-γ, TNF- α, Il-4, -5, -6, and -13, and chemokines (MCP-1, Il-8), which have a direct or indirect influence on aneurysm formation. In vitro and in vivo studies have proven that Il-6 affects the activation of lymphocytes, whereas IFN-γ stimulates cells such as macrophages, endothelial cells, lymphocytes, and fibroblasts. Likewise, Il-6 is necessary to determine the amount of lymphocytes and macrophages, as well as MMPs’ expression and its angiogenesis [60].

## 4. Potential Inflammatory Markers of AAA

AAA is an asymptomatic disease, and its pathogenesis is not sufficiently understood. Thus far, patients are diagnosed at an advanced stage of the disease, reporting, for example, pain in the chest area. In this case, the patient is treated surgically because no drug treatment has been developed that is effective in treating AAA. These operations are life-threating. Therefore, the search for and identification of inflammatory markers are of great importance in the rapid diagnosis of the disease, but also contribute to the knowledge and understanding of AAA’s pathogenesis.

There are several of the most common blood biomarkers that are related to the aneurysm formation process. Table 3 summarizes types of AAA markers and their role in AAA development (Table 3).

### 4.1. Cathepsin

Cathepsins (Cat) are the first group of biomarkers to be described. Until now, 11 types of Cat have been identified which occur in humans, including Cat: B, C, F, H, K, L, O, S, V, W, and X. Cat are essentially a group of proteases because they are housekeeping enzymes. There are late endosomes and lysosomes, therefore, their activity presents acidic pH. pH regulates their activity but so do cystatins, which are considered one of the most important regulators. Cat are mainly lysosomal proteases, but in spite of this, they are found in the cytosol, cell membrane, and intercellular space [61].

This lysosomal protease is a key factor in vascular remodeling during the aneurysm formation process and plays significant role in cell signaling, cell apoptosis, antigen presentation, and T-cell activation [61,62,77]. Each type of cathepsin mentioned above has its own specific functions. Importantly, CatS plays a crucial role in angiogenesis by proangiogenic production of laminin 5 extracellular matrix peptides and the degradation of antiangiogenic peptides that results from collagen IV proteolysis. These processes indicate the potential participation of cathepsin in the process of atherosclerosis, heart muscle remodeling, or allergic reactions. For this reason, Cat are believed to be involved in the development of cardiovascular disease. Disturbances in the ECM reconstruction process are the main cause of cardiovascular disease, and, in particular, these disorders are the cause of AAAs. A number of cytokines, such as TNF-α, which are inflammatory cytokines, and the binding of monocytes stimulate the expression and activity of cathepsins in ECs. These processes contribute to the initiation of AAA pathogenesis. Any disturbances resulting from the imbalance between endogenous cysteine cathepsin regulating inhibitors negatively affect the maintenance of the aortic wall structure, enhance aortic remodeling, and contribute to the formation of AAAs [61].

Studies have shown that levels of cathepsin K and S are low in healthy human arteries but are significantly increased in aneurysms that are formed. High levels of this Cat K and S have been found in macrophages, SMC, and ECs [61,78,79].

Studies on the role of cathepsins have also been carried out in murine models. Transgenic mice with a knockout of the cathepsin gene were examined. Three cathepsins were tested: CatS, CatK, and CatL. Angiotensin II infusion-induced mouse models lacking apolipoprotein E demonstrated increased expression of CatS in murine AAA. In contrast, the lack of cathepsin S reduced the incidence of AAA and diameter of the aorta. CatS deficiency improves arterial wall elastin integrity, collagen accumulation, and decreased expression of CatK and MMP-2 angiogenesis. The role of cathepsin K in AAA has been investigated in two models. The first model was a mouse model with a systemic infusion of Ang II Apoe−/−; the second used AAA perfusion-induced porcine pancreatic elastase in mice. The study showed that mice deficient in CatK developed smaller aneurysms 14 days after elastase perfusion. Importantly, the lack of CatK did not affect the macrophage content in the AAA lesion but decreased the content and proliferation of CD4^+^ T cells in the lesion as well as the overall change and apoptosis of SMC, which prevented the loss of SMC in the aortic wall. Research concluded that cathepsin K reduces the activity of CatL, MMP-2, and MMP-9. Cathepsin L has been tested in murine models. For this purpose, a mouse model used elastase perfusion-induced experimental AAAs and peri-aortic CaCl_2_ injury-induced aorta expansion. The absence of cathepsin L has been found to protect mice from AAA formation. In addition, the lack of CatL reduces the content of proinflammatory cells such as macrophages, but also angiogenesis, cell proliferation, and elastin degradation, but has no effect on the apoptosis of damaged cells [61,80,81].

To summarize, both human AAA studies and animal model studies show that these proteases may provide an important marker that will aid in the early diagnosis of patients with aneurysm (Figure 4).

### 4.2. Homocysteine

The second potential biomarker of AAA is homocysteine (Hcy). This biomarker is a nonprotein amino acid derived from the hepatic metabolism of methionine [72,73,74]. It is formed in all cells of the human body as an intermediate product along the methionine–cysteine pathway. The metabolism of homocysteine involves two main pathways: remethylation to methionine and transsulfuration to cystathionine and then to cysteine. In healthy people in states of methionine deficiency, homocysteine is mainly metabolized by remethylation to methionine [72,73].

The concentration of Hcy in the blood depends on many factors, for example: age, sex, the genetic background of the activity, and enzymes that are crucial for its transformation. Elevated blood homocysteine levels generally result from disturbed transsulfuration or remethylation processes. Food deficiencies in B vitamins affect the activity of enzymes for which they are cofactors and may negatively affect the metabolism of methionine. Maintaining the proper level of homocysteine is essential. If the amount of produced homocysteine exceeds the metabolic capacity of the cell, its excess is exported to the extracellular space. Homocysteine exhibits cytotoxic properties, and excessive accumulation in the blood has a detrimental effect, for instance, it causes endothelial damage, degradation of elastin in the intimal vascular membrane, fibrillation, and calcification processes. In the lens, homocysteine is oxidized to mixed disulfide, free homocysteine, or thiolactane cyclization. Thiolactane has the ability to acetylate all amino groups of proteins and, as a result, they lose their biological activity [72,73].

Due to the cytotoxic properties of homocysteine and the consequences resulting from its action, it is considered as a risk factor for cardiovascular diseases, including abdominal aortic aneurysm. Determining the concentration of this amino acid in the blood will contribute to earlier prophylaxis as well as effective therapeutic activities [72,73,74,75].

It is now believed that the degradation of the extracellular matrix by the action of specialized proteolytic enzymes is one of the key steps in the early formation of an abdominal aortic aneurysm. In the aortic wall, proteolytic processes occur in which enzymes such as a tissue plasminogen activator or plasmin and a number of metalloproteinases (e.g., MMP-2 and MMP-9) are involved.

### 4.3. Osteoprotegerin

Scientists conclude that osteoprotegerin (OPG) is one of the key markers that may contribute to the prompt diagnosis of patients with AAA [67]. OPG is considered as a key regulator of bone homeostasis. It is secreted by a number of cells, for instance, smooth muscle cells and the endothelium. Endothelial cells secrete OPG but also become subject to the action of this glycoprotein. OPG belongs to the TNF-α receptor superfamily and is a glycoprotein cytokine receptor. TNF-α is an integral part of the inflammatory process in AAA. Elevated TNF levels correlate with the number of inflammatory cells involved in the inflammatory process and the degree of vascularization. However, OPG is not only associated with the formation of AAA, but also with the formation of atherosclerosis, endothelial damage, valvular heart disease, and peripheral arterial disease [67]. It is known that in the wall of the abdominal aortic aneurysm, an inflammatory process develops where various proinflammatory cytokines promote the activation of proteolytic enzymes, including TNF-α, OPG, and osteopontin (OPN). An increased concentration of OPG affects the initiation of AAA. The assessment of plasma OPG concentration is crucial in assessing patients with atherosclerosis and AAA. OPG is expressed in vascular walls, endothelial cells, and VSMC. Pro-inflammatory interleukins such as TNFα and Il-1beta promote the release of OPG. Via monocytes, endothelial cells, and VSMC, OPG stimulates the secretion of metalloproteinases such as MMP-2 and MMP-9. These metalloproteinases are crucial in the correct remodeling of the ECM; their impaired expression affects the degradation of the ECM and, consequently, the destruction of the aortic wall. The correlation of OPG concentration with the aneurysm diameter was also examined. Aneurysm diameter has been shown to influence the level of OPG concentration, and it is associated with large aneurysms [67,68].

The use of blood OPG concentration in routine diagnostics will contribute to faster diagnosis of patients with advanced patients who do not produce AAA symptoms.

### 4.4. Osteopontin

Osteopontin (OPN) is also a significant biomarker of AAA. It is a glycol-phosphoprotein that plays a role in the physiological and pathophysiological process, mainly responsible for the control of biomineralization and calcification. Under normal physiological conditions, the level of OPN is low, which contributes to the maintenance of blood vessel homeostasis. Low concentrations of OPN are critical for maintaining the mechanical properties of the arteries but also for the possibility of OPN acting as an inhibitor of vascular calcification. It has been shown that when tissue is damaged, the level of OPN rises sharply. High levels of OPN are beneficial for cell maintenance, agitation, proliferation, and migration, which promotes the wound healing process. Over time, the level of OPN expression decreases. Osteopontin is mainly secreted and regulated by ECs, VSMCs, and macrophages. If the OPN level does not drop, it indicates cardiovascular disease (CVD), but also autoimmune diseases or various types of cancer [65].

OPN is a characteristic factor in the inflammatory process. It interacts with proinflammatory cytokines such as IL1-β and TNF-α, which increase OPN expression, and with inflammatory mediators such as AngII, which are mainly associated with the atherosclerotic process. AngII is the major factor increasing the expression of OPN in the wall. It has also been shown that OPN is the main chemoattractant for cells of the immune system, for example, macrophages and T lymphocytes. OPN strengthens macrophages and helper activity of T lymphocytes, increasing the expression of CD3 by T lymphocytes. OPN contributes to the production of reactive oxygen species. Reactive forms of oxygen largely contribute to the formation of atherosclerotic plaques in the walls of the arteries, which promotes their degradation [66,69].

Characterization of the group of potential AAA biomarkers will enable early diagnosis of patients suffering from this disease. Early detection of AAA would allow for earlier treatment and significantly reduce the mortality rate in this group of patients.

## 5. Conclusions/Perspectives

AAA is a complex disease, and each case is slightly different [82]. This phenomenon is due to various factors such as environmental, genetic, and molecular. The degree of their influence on the process also affects AAA’s appearance. Extracellular matrix degradation and inflammation are considered to be the most important molecular processes contributing to AAA formation. The initiating molecular processes cause matrix degradation, cell apoptosis, and aging, as well as infiltration of inflammatory cells such as lymphocytes, mast cells, and neutrophils (Figure 5) [82,83,84].

To date, there is no effective pharmacological agent to treat AAA. Gene therapy, such as decoy therapy, is believed to be most effective, as multiple compensation pathways converge to activate a specific network of transcription factors [85].

Activation of transcription factors leads to the transcription of a set of genes associated with the pathological condition. Some of these molecules have the ability to activate transcription factors, resulting in the induction of a positive feedback loop, leading to the maintenance of the disease state. Importantly, a decoy strategy is available to regulate endogenous transcription factor activity [85,86].

Transcription factors mainly regulate the expression of pro-inflammatory factors such as cytokines and adhesion molecules. Nuclear factor-kappa B (NFκB) is a key transcription factor in both the acute and chronic inflammatory response of AAA. NFκB directly regulates many cytokines and proteases, such as Interleukin (IL)-1, IL-6, and tumor necrosis factor-α (TNF-α). Moreover, MMPs, TNF-α, and IL-1b can also activate NFκB. Studies have also shown that NFκB regulates the expression of adhesion molecules and chemokines that induce inflammatory cell migration [85].

Inflammatory cells, as well as mast cells, are the primary source of inflammatory cytokines and proteases. Inhibition of inflammatory cell recruitment indirectly suppresses the overexpression of inflammatory mediators. It has been shown that NFκB decoy ODN treatment mediated potent anti-inflammatory effects in rat and rabbit models of AAA. In addition, NFκB inhibited the transcription of elastin and collagen genes, suppressing their synthesis. This is why NFκB is believed to be the main target of decoy strategies for treating AAA [85,87,88].

## Figures and Tables

**Figure 1 ijms-23-11078-f001:**
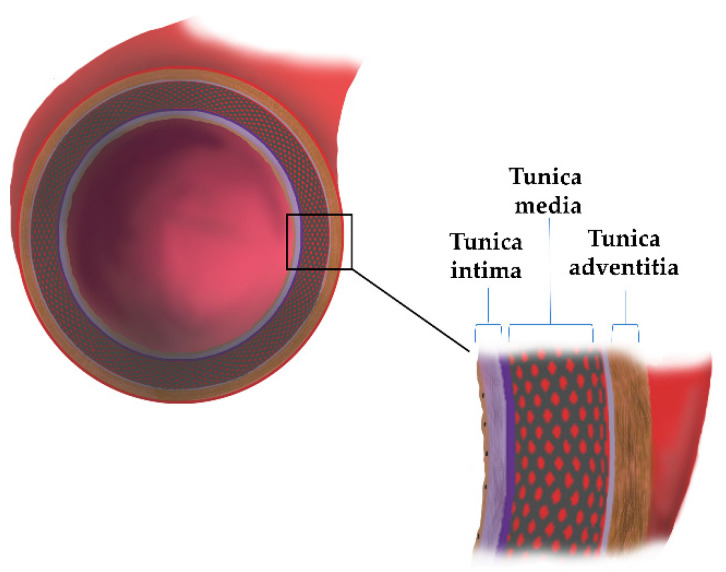
The schematic structure of the aorta wall.

**Figure 2 ijms-23-11078-f002:**
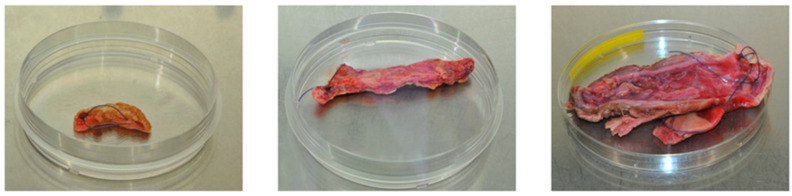
Abdominal aortic aneurysm samples obtained from patients after surgical treatment. Proximal part of the aneurysm is marked with stitches.

**Figure 3 ijms-23-11078-f003:**
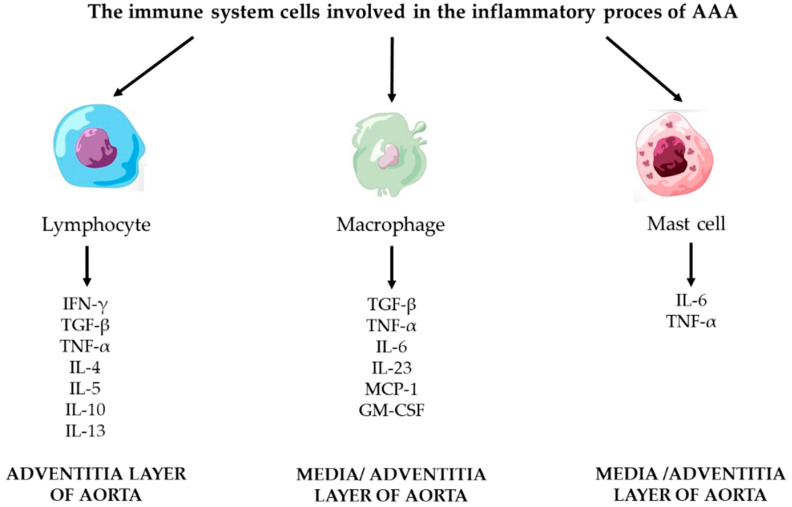
The immune system cells with mediators involved in the inflammatory process of AAA.

**Figure 4 ijms-23-11078-f004:**
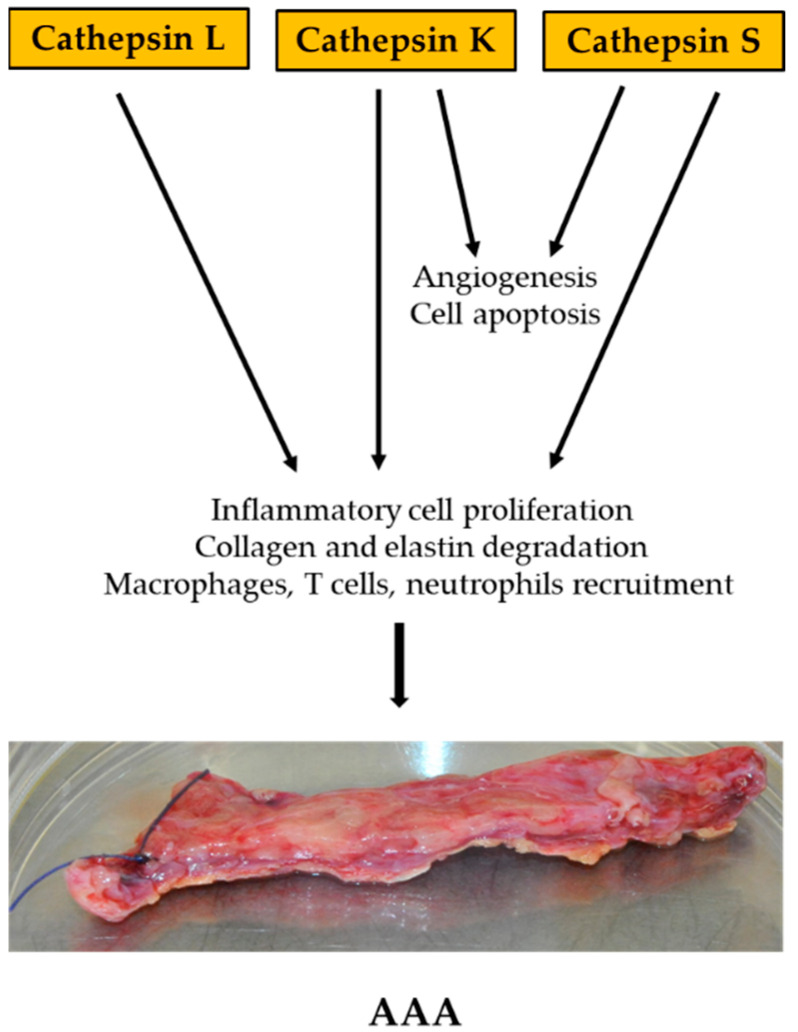
Diagram summarizing the role of cathepsins in AAA formation in vivo.

**Figure 5 ijms-23-11078-f005:**
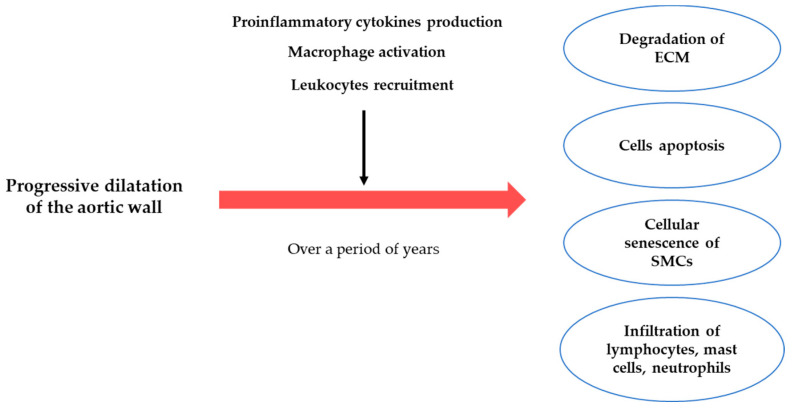
Schematic summarizing of molecular processes underlying abdominal aortic aneurysm formation.

**Table 1 ijms-23-11078-t001:** The aortic wall structure: tunica intima (inner layer), tunica media (medium layer), tunica adventitia (external layer).

Aortic Wall	Components	Functions	Reference
Tunica intima	Endothelial cells (ECs) Extracellular matrix network (ECM): laminin, collagen type IV, fibronectin, perlecan, heparan sulfate, proteoglycans, nidogen	Synthesis and release of inflammatory mediators, hormones, and factors that contract and relax arteries (NO, PGI2, E-selectin, ICAM-1)	[9,10,11]
Tunica media	Smooth muscle cells (SMCs); ECM proteins: Proteoglycans (PGs), Glycoproteins, Glycosaminoglycans (GAGs), collagen, elastin	Compliance and recoil properties	[9]
Tunica adventitia	Fibroblast (FBs), collagen type I and III, elastic fibers; chondroitin sulfate, dermatan sulfate Proteoglycans	Tensile strength	[9,11]

**Table 3 ijms-23-11078-t003:** A list of the most important markers involved in the formation of AAA.

Biomarker	Type of Molecule	Role in AAA	References
Cathepsin (Cat)	Protein	Vascular remodeling, cell apoptosis, cell signalling	[61,62]
Cystatin B, C	Protein	Aortic wall proteolysis contributing to AAA enlargement and rupture	[63,64]
MMP-2, -9	Protein	Degradation of ECM	[34,35]
Osteopontin (OPN)	Protein	Activation of immune cells; inflammation process	[65,66]
Osteoprotegerin (OPG)	Protein	Activation of immune cells; inflammation process	[67,68]
Thierodoxin (TRX)	Protein	Increases oxidative stress in the aorta wall	[69]
IL-1α	Protein	Inflammation proces	[47,50,51]
MiRNA712/205	Micro RNA	Enhances the secretion of MMP (MMP-3); development of inflammation; degradation of connective tissue and ECM	[52,70,71]
MiRNA 29C-3P	Micro RNA	Lowers the expression of genes encoding *ELN, COL4A1, VEGFA*	[70,71]
Homocysteine (Hcy)	Amino acid	Development of a blood clot in aneurysm, degradation of elastin in the inner membrane, fibrosis, and calcification processes	[72,73,74,75,76]

## Data Availability

Not applicable.

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
