# Peer review of "Role of Extracellular Matrix and Inflammation in Abdominal Aortic Aneurysm"

_ijms, 2022, doi:10.3390/ijms231911078_

Round 1
Reviewer 1 Report
In this review, the authors summarize the underlying molecular mechanisms and biomarkers of AAA. This review contributes to further understanding of AAA and development of new treatment strategies.
Comments,
1. Abstract, “Inflammation and degradation of the extracellular matrix in the AAA wall are believed to be the major molecular process underlying the AAA formation.” Depletion of aortic smooth muscle cells is also a main player in AAA formation.
2. There is a skip from “2.1.1. Types of fiber structures” to “3.2. Modifications of ECM structural elements.”
3. Too much introduction on “the aortic wall structure” and “ECM”, please rewrite and tailor this section. The topic is AAA, not aortic anatomy and physiology.
4. If AAA is used, avoid to appear “abdominal aortic aneurysms” again (line 254), check other parts. Other abbreviations, like TGF, also should be check.
5. Fig2, you cannot use previously published figs. Please confirm it is not same with cited papers and also fig4 in this review.
6. Show the + in like “CD4+” as upper, check similar problems.
7. Move macrophages to the front of Lymphocyte when discussed the Inflammatory process (line 297). So, the arrangement order is consistent with Fig 3.
8. Fig 3, macrophages were also found in adventitia.
9. In general, after discussion of the mechanisms of AAA, the authors should give a translational perspective of lab or clinical drugs, such as some anti-inflammatory molecules.
Author Response
- Abstract, “Inflammation and degradation of the extracellular matrix in the AAA wall are believed to be the major molecular process underlying the AAA formation.” Depletion of aortic smooth muscle cells is also a main player in AAA formation.
Line 17- we add : apoptosis of smooth muscle cells
- There is a skip from “2.1.1. Types of fiber structures” to “3.2. Modifications of ECM structural elements.”
The numbering of all chapters has been corrected.
- Too much introduction on “the aortic wall structure” and “ECM”, please rewrite and tailor this section. The topic is AAA, not aortic anatomy and physiology.
It has been changed in the mode of changes
- If AAA is used, avoid to appear “abdominal aortic aneurysms” again (line 254), check other parts. Other abbreviations, like TGF, also should be check.
It has been changed in the mode of changes
- Fig2, you cannot use previously published figs. Please confirm it is not same with cited papers and also fig4 in this review.
The placed images of aneurysm sections in our review paper are not previously published. The citation was added to show that in our earlier original papers, photos of aneurysm sections were placed.
- Show the + in like “CD4+” as upper, check similar problems.
It has been changed in the mode of changes
- Move macrophages to the front of Lymphocyte when discussed the Inflammatory process (line 297). So, the arrangement order is consistent with Fig 3.
It has been changed in the mode of changes - sequence of cells has been changed in the diagram
- Fig 3, macrophages were also found in adventitia.
We added it in the diagram
- In general, after discussion of the mechanisms of AAA, the authors should give a translational perspective of lab or clinical drugs, such as some anti-inflammatory molecules
A passage responding to suggestions was included in Chapter 5. Line 634 -655.
Reviewer 2 Report
I appreciate the authors for writing this review on the mechanisms of AAA and its prognostic biomarkers. Below are some of concerns/comments.
1. The review is largely focused on ECM and inflammatory markers. However, mechanisms of other characteristics including intraluminal thrombus (IGF, NGAL, peroxiredoxine), peak wall stress, and markers including telomere length, LPS, phospolipase, D-dimer, PAP, elastin peptides and TWEAK (to name a few) were not discussed.
2. Ideal title should be 'Role of ECM and inflammation in AAA'. Only a very few prognostic biomarkers are discussed, so it would be better not to use it in the title as several prognostic markers were not discussed.
3. Line 31 - AAA is not really common is general population, but slightly higher in those aged over 55. Instead of mentioning it as 'common', it would be better to provide recent prevalence data.
4. Line 37-40 - Provide reference. In line 39, the word 'medium' should be defined clearly with more precise diameter size.
5. Line 55-56 - Rephrase the sentence for clarity.
6. Throughout the paper, an overall review is cited as reference at the end of the paragraph in many subsections instead of citing reference of each mentioned point. This would be hard for the readers to follow in case of citing specific points.
7. Some evidences mentioned about AAA mechanism cite different conditions and animal models/human condition such as pressure overload mice (ref 16), obese women (ref 18), etc but this is not clearly mentioned in the text. Each evidence/points should also mention which animal model or human condition that evidence came from. Also, evidences provided are mixed between animals/humans which reduces its reliability.
8. Table 2 - The column 'Role in pathogenesis of AAA' is rather speculative with the references provided. For example, ref 37 and 41 is an in vitro study, so cannot be reported as pathogenesis of AAA. ref 38 is a computer modeling data, ref 39 is from ASD patients. It would be better to rename this column as 'potential role in AAA pathogenesis'. Also, include a column mentioning what type of tissue/in vivo model/human condition the study was performed.
9. Check all references and if the study is not done in AAA patients/model, please mention it in the text.
Author Response
1.The review is largely focused on ECM and inflammatory markers. However, mechanisms of other characteristics including intraluminal thrombus (IGF, NGAL, peroxiredoxine), peak wall stress, and markers including telomere length, LPS, phospolipase, D-dimer, PAP, elastin peptides and TWEAK (to name a few) were not discussed.
Thank you very much for the comment, but the suggestions in it are not the subject of the paper. On the other hand, of course, they are relevant and will certainly be discussed in our next work.
- Ideal title should be 'Role of ECM and inflammation in AAA'. Only a very few prognostic biomarkers are discussed, so it would be better not to use it in the title as several prognostic markers were not discussed.
I appreciate this proposal – title has been changed.
- Line 31 - AAA is not really common is general population, but slightly higher in those aged over 55. Instead of mentioning it as 'common', it would be better to provide recent prevalence data.
Modified by changes
- Line 37-40 - Provide reference. In line 39, the word 'medium' should be defined clearly with more precise diameter size.
Thank you for this observation, the information has been checked with sources and is adequate.
- Line 55-56 - Rephrase the sentence for clarity.
Modified by changes.
- Throughout the paper, an overall review is cited as reference at the end of the paragraph in many subsections instead of citing reference of each mentioned point. This would be hard for the readers to follow in case of citing specific points.
Citations have been clarified - Modified by changes.
- Some evidences mentioned about AAA mechanism cite different conditions and animal models/human condition such as pressure overload mice (ref 16), obese women (ref 18), etc but this is not clearly mentioned in the text. Each evidence/points should also mention which animal model or human condition that evidence came from. Also, evidences provided are mixed between animals/humans which reduces its reliability.
Corrections were made in the mode of changes. Line 349, 352, 353
- Table 2 - The column 'Role in pathogenesis of AAA' is rather speculative with the references provided. For example, ref 37 and 41 is an in vitro study, so cannot be reported as pathogenesis of AAA. ref 38 is a computer modeling data, ref 39 is from ASD patients. It would be better to rename this column as 'potential role in AAA pathogenesis'. Also, include a column mentioning what type of tissue/in vivo model/human condition the study was performed.
the name of the column has been changed.
- Check all references and if the study is not done in AAA patients/model, please mention it in the text.
References were checked and corrections were made
Round 2
Reviewer 2 Report
Thanks for getting back with the revision. I see that a couple of comments were not clearly addressed. Please see below.
1. In comment 3, most recent prevalence data was requested. I could not see this point addressed.
2. In comment 4, the authors responded that the information provided is adequate. I am sorry I could not verify the term 'medium diameter' anywhere in published studies. Did the authors meant to say 'mean diameter'? Please clarify.
3. In comment 7, I had queried about reference 16 and 18 but authors have made the changes in lines 349, 352 and 353. I am not sure how this response is relevant to the question.
Author Response
- In comment 3, most recent prevalence data was requested. I could not see this point addressed.
Response to Comment 3: We would like to kindly emphasize that the word “common” has been removed from the reviewed article (line 32). Nevertheless, in literature AAA is described as: “ (…) AAA is detected via ultrasonography screening programmes in women and men are available from only a few countries. (…)”. Nowadays, the knowledge about molecular aspects of AAA formation is still unclear. Differences between sizes and structure of AAA in patients are heterogeneous. We have shown such changes on Figure 2 (line 265).
Interestingly, the word “common” has been also used for description of AAA in article Sakalihasan [ Sakalihasan N, Michel JB, Katsargyris A, Kuivaniemi H, Defraigne JO, Nchimi A, Powell JT, Yoshimura K, Hultgren R. Abdominal aortic aneurysms. Nat Rev Dis Primers. 2018 Oct 18;4(1):34. doi: 10.1038/s41572-018-0030-7]. The crucial thing is that AAA’s rupture depends on many various factors but unfortunately 50% of patients die before hospitalisation. In general, the mortality of such patients is high because of occurrence atherosclerotic disease (i.e. cardiovascular disease or different types of cancer), as well as of ethnics group, age, sex and life style at the same time.
- In comment 4, the authors responded that the information provided is adequate. I am sorry I could not verify the term 'medium diameter' anywhere in published studies. Did the authors meant to say 'mean diameter'? Please clarify.
Response to Comment 4: We have collected many various AAA in terms of: size, diameter, structure. The word “medium” refers to the average value of AAA’s diameter. We apologize for misunderstanding but kindly ask for acceptation of this explanation. We would like to emphasize in the reviewed manuscript that diameter of AAAs may be varied depending on patient. We have seen diameter values in the range of 50 – 60 mm in our analysis.
- In comment 7, I had queried about reference 16 and 18 but authors have made the changes in lines 349, 352 and 353. I am not sure how this response is relevant to the question.
Response to Comment 7: Changes in line 349, 352 and 353 has been introduced because we would like to clarify description of models (animal and human) that have been ever used for AAA studies.
We apologize for missing the answer for this comment. Please see completed information in line 163-164 and 171.
We hope for positive consideration of our Response.